# Holistic Advances through Large-Scale Embodied Dialog Generation for Navigation

## Abstract

For embodied agents capable of physical interaction, dialog capability is crucial to ensure both safety and effectiveness. While DialNav provides a framework for holistic evaluation of the dialog–execution loop in photorealistic indoor navigation, its performance is constrained. In this work, we introduce holistic advances spanning data and training. First, we develop a large-scale dialog generation pipeline to enhance coverage and diversity. Second, we propose task-aligned training for the Navigator to better reflect the dynamic dialog–navigation loop. Finally, we address the bottleneck of localization with a stronger graph-aware transformer model. Together, these advances more than double success rates over prior baselines, achieving 58.24% SR on *Val Seen* and 29.05% on *Val Unseen*, establishing a new state of the art in dialog-driven embodied navigation.

## 1 Introduction

Embodied agents must operate with high reliability, since misinterpreting instructions or executing incorrect actions can cause physical harm. Enabling dialog improves both safety and task effectiveness: by asking questions and refining their understanding before acting, agents can resolve ambiguity and adapt to dynamic situations. Learning dialog-enabled agents, however, remains highly challenging. Collecting datasets is costly, as it requires two people to coordinate in real time while grounding their conversations in the given task context. Furthermore, even with such data, training remains difficult because if either the action trajectory or the dialog deviates from the collected annotations, the supervision becomes invalid.

We study these challenges in **DialNav** (Han et al., 2025), a cooperative dialog-based vision-and-language navigation (VLN) (Anderson et al., 2018) task. VLN is an embodied navigation problem in which an agent follows a fixed natural language instruction given at the start of an episode to reach a goal in a photo-realistic environment. DialNav extends this into a dialog-based setup, where navigation unfolds through an interactive dialog exchange. In DialNav, the Navigator receives only a coarse initial description (e.g., "the target room has a carpet"), which is insufficient for reaching the goal, success therefore hinges on multi-turn interaction. While DialNav is a valuable benchmark, its practical use has been limited. The accompanying RAIN (Han et al., 2025) dataset contains only 2K episodes, which is too small to capture the complexity of dialog-conditioned navigation. Furthermore, existing training strategies largely target a static instruction setup, failing to reflect the interdependence of dialog and navigation at test time. Finally, localization—the Guide's ability to infer the Navigator's position from dialog—remains underexplored, despite being a key bottleneck that propagates errors throughout the entire collaboration loop.

To address these limitations, we introduce three complementary advances. First, we develop an automatic pipeline that converts existing fine-grained VLN resources into dialog-style navigation episodes, yielding a corpus more than two orders of magnitude larger than the original RAIN dataset. Second, we design history and interaction aware navigator training tailored for multi-turn dialog. Finally, we tackle the localization bottleneck with a graph-aware transformer module that significantly reduces mislocalization and its cascading errors. Together, these advances more than double navigation performance: success rate rises from 28.57 to 58.24 on *Val Seen* and from 13.69 to 29.05 on *Val Unseen*.

In summary, our contributions are threefold:

- We generate DialNav-style data without human annotation, constructing a corpus over two orders of magnitude larger than the original RAIN dataset.

- We enhanced training strategy for DialNav, which faithfully capture the interactive dialog–action loop while substantially reducing mislocalization.

- Our integrated system establishes a new state of the art in dialog-driven navigation, more than doubling success over the prior baseline

## 2 RELATED WORK

**Vision and Language Navigation.**    Visual-and-Language Navigation (VLN) is an embodied AI task in which an agent navigates through visual environments by following natural language instructions. VLN was initially explored under a static instruction setting, where the instruction is provided once at the beginning of navigation and remains unchanged throughout the episode. Instructions are either coarse-grained (Zhu et al., 2021; Qi et al., 2020), often too ambiguous for reliable navigation, or fine-grained (Anderson et al., 2018; Ku et al., 2020), often overly detailed and unnatural for human interaction. More recently, dialog-based approaches have been introduced. CVDN (Thomason et al., 2020) collected human–human dialog data for navigation, but this has mostly been studied under the Navigation from Dialog History (NDH) setup, in which the segmented dialog is provided as an instruction at the corresponding node to guide the agent, failing to capture the full dynamics of multi-turn dialog in navigation. To address this gap, DialNav (Han et al., 2025) was proposed, explicitly emphasizing the role of dialog by formulating navigation as a dialog–navigation loop that holistically integrates subtasks including question–answer generation (Fried et al., 2018; Fan et al., 2023b; Chen et al., 2022a), localization (Hahn et al., 2020; Wang et al., 2025; Zhang et al., 2024; Hahn & Rehg, 2022; Pate & Wong, 2023; Li et al., 2024), and navigation (Anderson et al., 2018; Chen et al., 2022b; 2021). Nevertheless, this setting remains under-explored: the dataset is relatively small, and the integrated formulation poses significant challenges, resulting in low performance. In this work, we tackle these challenges by introducing dialog-specific augmentation and dialog-aware navigator training, substantially improving performance in dialog-based VLN.

**Embodied Dialog Dataset and Augmentation.**    Dialog-based formulations have been explored across multiple embodied tasks, including Embodied Question Answering (Das et al., 2018; Majumdar et al., 2024), Vision-and-Dialog Navigation (Thomason et al., 2020; Fan et al., 2023a; de Vries et al., 2018; Han et al., 2025), and embodied manipulation (Gao et al., 2022; Padmakumar et al., 2022). These tasks are predominantly based on fully human-annotated dialogs, which makes data collection costly and limits scalability to larger datasets. Recent efforts have begun to explore dialog augmentation in embodied settings including leveraging large language models (Zhang et al., 2024) or using template-based procedure (Padmakumar et al., 2023). However, these methods remain superficial or restrict diversity and realism. We propose a multi-turn dialog generation for vision-and-dialog navigation that expands training episodes by leveraging fine-grained VLN datasets.

## 3 PRELIMINARIES

**DialNav Task.**    DialNav (Han et al., 2025) is a dialog-based navigation task, where a mobile agent (*Navigator*) moves through a photo-realistic indoor environment Matterport3D (Chang et al., 2017) to reach a goal region with assistance from a *Remote Guide*. At the beginning of a DialNav episode, the navigator is given **only** a high-level initial instruction (*e.g.*, "the target room has a carpet"), which alone is insufficient for successful navigation. Success therefore hinges on dialog: through multi-turn exchanges, the Guide provides essential information that enables the Navigator to locate and navigate toward the target. The overall DialNav task is illustrated in Figure 1 (top).

**Notation.**    Formally, the environment in the DialNav task is represented with a connectivity graph $G = (V, E)$ where $v \in V$ is navigable node, $E \subseteq V \times V$ encodes adjacency between the adjacent nodes. A DialNav episode with the $K$ steps is $\mathcal{E} = (G, b, R, I, T_K, D_K)$ where $b \in V$ is the initial node, $R \subseteq V$ is the goal region spanning multiple adjacent nodes, and $I$ is an initial instruction. $T_K = (v_0 = b, \ldots, v_K)$ denotes the navigation trajectory accumulated up to the step $K$, which is a sequence of nodes that the Navigator has traversed. The dialog $D_K = \{(v, q, a)\}$ is a cumulative set of a triplet of a question $q$ posed at node $v$ and its corresponding answer $a$, up to the step $K$.

**Navigator and Guide Workflow.**    The DialNav episode proceeds in rounds, as illustrated in Figure 1 (bottom). At each step $t$, the Navigator first decides whether to request assistance, denoted as

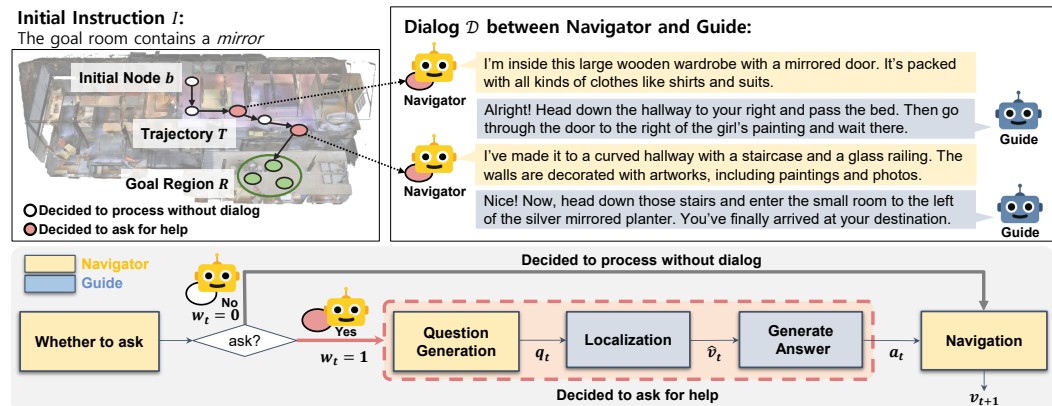

Figure 1: **Overview of the DialNav task**. **Top:** The Navigator starts at an initial node $b$ and must reach a goal region $R$ in a photo-realistic indoor environment. Since the initial instruction is under-specified, the Navigator engages in multi-turn dialog with a Remote Guide to successfully reach the target region. **Bottom:** At each step, the Navigator follows a modular decision process: it either proceeds independently (*Navigation*) or requests help (*Question Generation*). When a question is asked, the Guide localizes the Navigator and provides an answer describing the next path to the goal. This forms an alternating loop of dialog and action that continues until the goal region is reached.

$w_t$. If the Navigator chooses to proceed without assistance, *i.e.*, $w_t = 0$, the Navigator advances autonomously to the next viewpoint $v_{t+1}$. If the Navigator requests help, *i.e.*, $w_t = 1$, the Navigator issues a question $q_t$. Upon receiving the query, the Guide infers the Navigator's location $\hat{v}_t$ and provides an answer $a_t$, typically describing an optimal path from the inferred position $\hat{v}_t$ toward the goal $R$. Then, the Navigator continues navigation using this answer as a guidance until a new conversation between the agents occurs. This process repeats until the Navigator determines that it has reached the goal region $R$.

**Training with Segmented Data.**    A natural approach to implementing the two agents for the DialNav task is to decompose their core tasks into distinct networks: the *Navigator* comprises navigation, question generation, and whether-to-ask modules, while the *Guide* comprises localization and answer generation modules. To the best of our knowledge, the only prior work that has instantiated this approach in DialNav is Han et al. (2025), who trained each module independently by adapting existing models from related areas Because these models were originally designed for static contexts without multi-turn interaction, Han et al. (2025) partitioned the dialog data into training segments that serves as supervision for each module. A segment is defined as $\mathcal{S}_j = (v_j, D_j)$, where $v_j$ denotes the node at the $j$-th step and $D_j$ is the dialog history accumulated up to that point. Each segment then becomes a static instruction-following instance: the question generation module predicts $q_j$, the localization module predicts $v_j$, and the answer generation module predicts $a_j$. For navigation, the segment corresponds to navigating from $v_j$ to the goal region $R$, conditioned on the initial instruction $I$ and dialog history $D_j$.

**Limitation.**    While prior research on DialNav (Han et al., 2025) has demonstrated the feasibility of training agents for this task, several limitations remain that hinder high performance. First, Han et al. (2025) relied on the RAIN dataset, which was directly collected from human annotators. However, its scale (∼2K episodes) is too limited to serve as sufficient training data for a task as complex as DialNav. Second, the two agents are composed of separate neural modules that are trained independently under static settings, without modeling the interaction between agents. When deployed in a holistic dialog–navigation loop, this mismatch causes a noticeable drop in overall performance. Finally, localization remains relatively underexplored and continues to be a key bottleneck that constrains task success in DialNav.

## 4    METHOD

To address the aforementioned limitations, we propose two complementary advances. First, we develop a *human-annotation-free* data generation pipeline, expanding the training corpus by over 100 times to the original RAIN dataset. Second, we design training schemes that explicitly capture the evolving dialog and the remote guidance setting of DialNav.

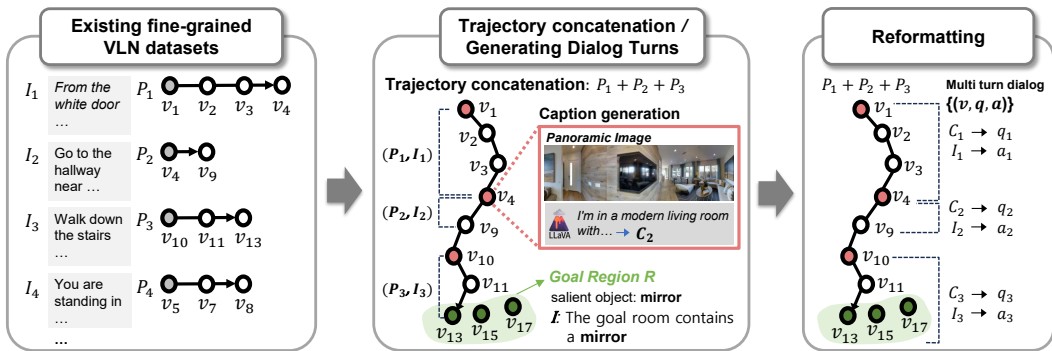

Figure 2: **Overview of the dataset generation pipeline.** (Left) We start from existing fine-grained VLN datasets, where each path $P_i$ is paired with a fine-grained instruction $I_i$. (Middle) Multiple paths are concatenated into an extended trajectory, and at the starting node of each constituent path, a panoramic caption $C_i$ is generated using a vision–language model. The original fine-grained instructions $I_i$ are repurposed as dialog answers, while the final node defines the goal region $R$. (Right) Caption–instruction pairs $(C_i, I_i)$ are then reformatted into natural multi-turn dialogs using a large language model, producing large-scale dialog-style data for DialNav training.

## 4.1 AUTOMATIC DIALNAV EPISODE GENERATION

We present a novel pipeline that automatically generates DialNav-style data from fine-grained VLN instruction datasets, where each data instance consists of a navigation path and its associated fine-grained instruction. Specifically, we *concatenate* existing paths from VLN datasets to form new trajectories, each with an initial node and a goal region. To enable multi-turn dialog, we regard the starting node of each individual path as a new conversation point between the Navigator and the Guide (*i.e.*, where a question–answer exchange begins). In other words, when multiple paths are concatenated into a longer trajectory, the starting node of each constituent path serves as the entry point for generating dialog turns. For question generation, we prompt a large vision–language model to *caption* the node corresponding to the conversation point based on its panoramic observation. For answer generation, we pair the fine-grained instruction corresponding to the selected path with a language model to produce a natural response. Finally, we *reformat* these captions and instructions into coherent multi-turn dialogs using an LLM. Through this process, we construct a large-scale corpus of dialog episodes that is over 100 times larger than the existing RAIN dataset (Han et al., 2025). Our overall generation pipeline is illustrated in Fig. 2.

**Trajectory Concatenation.** We first build extended navigation trajectories by concatenating multiple path–instruction pairs $\{(P_i, I_i)\}$ from fine-grained VLN instruction datasets such as R2R (Anderson et al., 2018) and RxR (Ku et al., 2020). The resulting concatenated path $T$ has an initial node $b$, a goal region $R$ (the room containing the final node, following metadata partitions (Chang et al., 2017)), and a navigation trajectory spanning all intermediate viewpoints. The initial instruction $I$ is then derived by randomly selecting a salient object within $R$, also based on the metadata. For dialog construction, we treat the *starting node of each constituent path $P_i$* (before concatenation) as the point where a new conversation turn begins.

**Generating Dialog Turns.** At each such starting node, the Navigator poses a question and the Guide provides an answer. To obtain natural questions, we caption the panoramic observation of the node using a large vision–language model (e.g., LLaVA-1.5-7B (Liu et al., 2023)). For answers, we repurpose the original fine-grained instruction $I_i$ associated with the path. This yields paired captions and instructions $(q_i, a_i)$ that simulate question–answer exchanges grounded in the environment. Details of the captioning prompts and implementation are provided in the Supp. Mat. E.

**Reformatting into Coherent Multi-turn Dialog.** Finally, we reformat the caption–instruction pairs obtained from the previous steps into coherent multi-turn dialogs. While the instruction $I_i$ describes a subpath in an imperative style and the caption $C_i$ provides visually grounded descriptions of nodes, dialogs in DialNav require natural conversational conventions. To bridge this gap, we employ a large language model (e.g., GPT-4o-mini (OpenAI, 2024)) to rewrite the raw $(I_i, C_i)$ pairs into fluent question–answer exchanges $(q_i, a_i)$. Through this process, we obtain large-scale dialog trajectories that are over two orders of magnitude larger than the existing RAIN dataset (Han et al.,

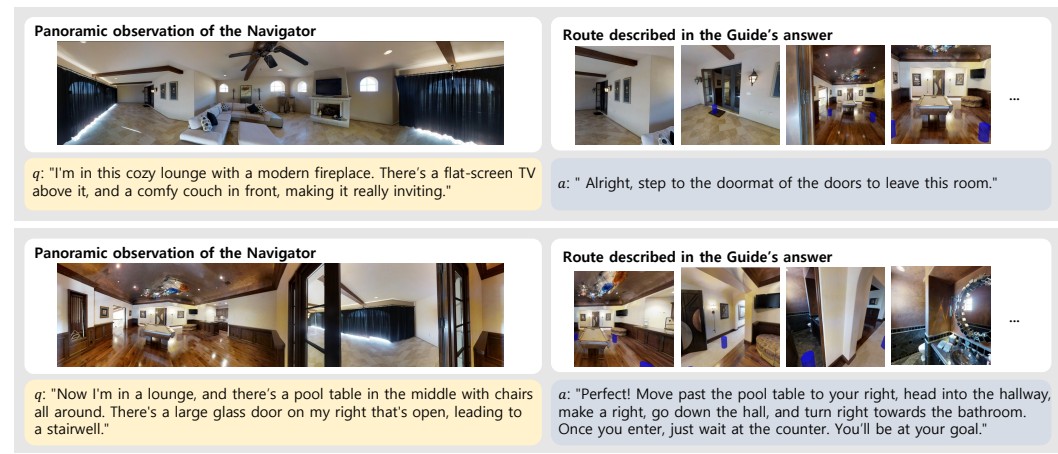

Figure 3: **Qualitative examples of grounded synthetic dialogs.** Generated questions are closely aligned with the Navigator's panoramic observations (left), and the Guide's answers provide route instructions that are visually consistent with the depicted environment (right). These examples illustrate that the automatically generated dialogs are not only fluent but also well-grounded in the underlying visual and spatial context.

2025). Figure 3 shows an example of the final generated dialog data. Details of the prompting and more examples of reformat output are provided in the Supp. Mat. F, A.

## 4.2 Enhanced Training Strategy for DialNav

To implement Navigator and Guide agents for the DialNav task, Han et al. (2025) adopted a modular design that decomposes the agents' core capabilities into distinct modules, drawing models from related subtasks that were originally studied in isolation. While we follow this design, we extend it by introducing navigation training schemes tailored for DialNav and by developing a stronger localization model, thereby enhancing the agents' overall performance in interactive navigation.

### 4.2.1 History and interaction aware navigator training

A critical challenge in DialNav lies in handling navigation under dynamically evolving dialog. Existing methods (Han et al., 2025) train the navigator following the conventional VLN paradigm, where the agent receives a single instruction at the initial node and is subsequently forced to follow the shortest path to the goal. Specifically, at the midpoint of the trajectory where dialog took place, the entire dialog history is concatenated into a single long instruction, and the navigator is tasked to reach the goal by following the shortest path while assuming that no further interactions will take place. While convenient, this approximation disregards the distinctive characteristics of DialNav: the agent's trajectory unfolds alongside dialog exchanges, and both past interactions and potential future instructions critically shape the navigation process.

To address these limitations, we introduce two training schemes explicitly designed for DialNav: (1) *model state building* enables the agent to condition on accumulated navigation–dialog history, ensuring that prior context is effectively incorporated into decision-making; (2) *decoupled episodic training* aligns navigation and dialog temporally by enforcing consistency between dialog turns and episode rollouts.

**Model State Building with History.** We extend the conventional VLN training paradigm by initializing the model's internal state directly from the accumulated history of dialog and navigation. Concretely, given a DialNav episode, we begin at the initial node and roll out the full annotated trajectory up to the midpoint corresponding to the starting node of the current training sample. This procedure embeds both past dialog exchanges and navigation trajectory into the model's hidden state. Conditioned on this history-aware state, the model is then trained to follow the shortest path to the goal. Although the training objective remains consistent with standard VLN, this formulation

preserves the temporal context of navigation and dialog, ensuring that the learning process is better aligned with the cumulative histories encountered at test time.

**Decoupled Episodic Training.** While model state building incorporates dialog and navigation history into the training pipeline, it overlooks the possibility of future interactions. The difficulty of accounting for future interactions arises from the fact that annotated dialogs are valid only under the exact past trajectory. Conversely, training the agent to strictly follow the annotated trajectory is also suboptimal, as human annotators may occasionally misinterpret instructions and navigate along suboptimal paths. To address this dilemma, we decouple the rollout trajectory from the training signal. Specifically, the agent is constrained to execute the annotated trajectory during rollouts, ensuring alignment with the available dialog annotations. At each node, however, the model is supervised to align with the shortest path, thereby learning locally optimal navigation decisions. Formally, the loss is defined as:

$$\mathcal{L}_{\text{dep}} = -\sum_{t=1}^{K-1} \log p(\hat{v}_t | v_{1:t}, D_t), \tag{1}$$

where $v_{1:t}$ denotes the partial trajectory from timestep 1 to $t$ of the annotated trajectory $T$, and $\hat{v}_t$ is the immediate next node on the shortest path from the last visited node $v_t$ to the goal. This formulation ensures that, on the one hand, the agent learns to make locally optimal navigation decisions at every visited node. On the other hand, the rollout trajectory remains aligned with the annotated data, enabling the model to fully leverage dialog supervision. By decoupling the learning signal from the rollout path, we achieve a balance between optimal navigation learning and faithful utilization of annotated dialogs.

### 4.2.2 GRAPH-AWARE TRANSFORMER BASED LOCALIZATION

In DialNav, localization errors cascade through the dialog–action loop: once the Navigator is mislocalized at a turn, the Guide's answer is conditioned on an incorrect pose and the subsequent navigator must reason from the corrupted answer. Compared to our baseline learned localization module GCN, which simply aligns LSTM-encoded text vector with mean-pooled panoramic node features, replacing localization with *ground-truth* poses yields a large performance gain (*cf.*, Tab. 4), indicating that localization is a primary bottleneck in the DialNav system. To mitigate this bottleneck, we use the graph-aware transformer architecture motivated by **DUET** (Chen et al., 2022b), which is well-known for a strong VLN architecture.

Formally, we first encode node representation $X'_v$ of each node $v$ from their panoramic visual observation $X_v$. Given this encoded representation and the question embedding $q'$, we compute a localization score for each candidate node $v \in G$ as:

$$s_v^{\text{loc}} = \text{FFN}(\text{GASA}(X'_v, q')), \tag{2}$$

where GASA denotes the graph-aware self-attention layer adapted from DUET. The model is trained using cross-entropy loss over all candidate nodes:

$$\mathcal{L}_{\text{loc}} = -\sum_{v \in V} y_v \log \frac{\exp(s_v^{\text{loc}})}{\sum_{u \in V} \exp(s_u^{\text{loc}})}, \tag{3}$$

with $y_v$ as the one-hot indicator of the true node.

Since our localization module shares its architecture with DUET, we can directly leverage publicly available weights pretrained on large-scale VLN datasets (Wang et al., 2023b) for initialization.

## 5 EXPERIMENTS

### 5.1 EXPERIMENTAL SETTINGS

**Models and Modules.** Following the modular design of DialNav, we train Navigator and Guide with separate submodules. Navigation adopts DUET (Chen et al., 2022b) as the backbone; question/answer generation uses LANA (Wang et al., 2023a); For localization, we compare GCN, the baseline model from prior work (Han et al., 2025) with our graph-aware transformer model. For all

Table 1: **Performance comparison across adopted methods.** DA: Dataset augmentation. TC: Training change. LC: Localization change. Performance improves as more components are combined, with all three enabled achieving the best results on both seen and unseen splits.

| Setup | Val Seen | | | | | | | Val Unseen | | | | | | |
|---|---|---|---|---|---|---|---|---|---|---|---|---|---|---|
| | SR↑ | OSR↑ | SPL↑ | NE↓ | NSC | DTC | LE↓ | SR↑ | OSR↑ | SPL↑ | NE↓ | NSC | DTC | LE↓ |
| Baseline (Han et al., 2025) | 28.57 | 36.26 | 26.64 | 11.06 | 18.75 | 4.59 | 16.95 | 13.69 | 17.84 | 10.15 | 18.62 | 26.16 | 13.98 | 23.83 |
| +DA | 27.47 | 39.56 | 25.24 | 10.72 | 20.40 | 8.00 | 16.00 | 17.43 | 21.58 | 11.89 | 17.16 | 28.05 | 17.80 | 19.82 |
| +DA+TC | 42.86 | 52.75 | 38.29 | 8.41 | 21.73 | 7.12 | 16.51 | 19.92 | 26.14 | 14.42 | 16.75 | 27.00 | 15.95 | 21.45 |
| +DA+LC | 32.97 | 47.25 | 30.76 | 9.26 | 19.85 | 5.76 | 4.93 | 22.82 | 27.80 | 17.00 | **13.18** | 20.89 | 10.57 | 9.75 |
| **+DA+TC+LC (Ours)** | **58.24** | **68.13** | **51.65** | **5.38** | 21.30 | 4.66 | 5.51 | **29.05** | **31.54** | **19.61** | 13.91 | 21.46 | 11.08 | 13.07 |

Table 2: **Dataset augmentation ablations.** RAIN: training only with the RAIN dataset. + Fine-grained inst.: adding VLN fine-grained instructions. + Traj. Concat: concatenating trajectories from fine-grained instructions. + Dialog Turns: generating dialog turns from concatenated trajectories. + Reformat: reformatting into natural multi-turn QA dialogs. Enhance training strategies described in §4.2 are fully adopted across all dataset variations.

| Dataset | Val Seen | | | | | | Val Unseen | | | | | |
|---|---|---|---|---|---|---|---|---|---|---|---|---|
| | SR↑ | OSR↑ | SPL↑ | NE↓ | NSC | DTC | SR↑ | OSR↑ | SPL↑ | NE↓ | NSC | DTC |
| RAIN | 37.36 | 47.25 | 33.74 | 8.54 | 22.16 | 6.01 | 23.24 | 29.46 | 16.49 | 14.68 | 20.80 | 8.85 |
| + Fine-grained inst. | 40.66 | 51.65 | 32.73 | 8.94 | 22.93 | 9.56 | 25.73 | **34.44** | 17.92 | 14.37 | 24.68 | 12.67 |
| + Traj. Concat | 39.56 | 48.35 | 35.36 | 7.46 | 20.54 | 4.00 | 19.09 | 27.80 | 13.79 | 17.04 | 25.8 | 13.06 |
| + Dialog Turns | 56.04 | 65.93 | 51.41 | 5.51 | 20.22 | 5.80 | 22.82 | **34.44** | 15.47 | 14.32 | 23.15 | 12.19 |
| **+ Reformat (Ours)** | **58.24** | **68.13** | **51.65** | **5.38** | 21.30 | 4.66 | **29.05** | 31.54 | **19.61** | **13.91** | 21.46 | 11.08 |

models, we leveraged publicly available pretrained weights prior. When training with augmented data we build from 4.1, we mix RAIN:Aug at a 1:9 ratio.

**Evaluation.** We evaluate performance on RAIN under the holistic dialog–navigation loop setting. Results are reported on the *Val Seen* split, consisting of environments overlapping with training, and the *Val Unseen* split, consisting of novel environments (Anderson et al., 2018). For the whether-to-ask (WTA) decision, we fix the confidence threshold at 0.9, so that the agent asks a question whenever its next-action confidence falls below this value. Performance is measured using standard VLN metrics: Success Rate (SR), Oracle Success Rate (OSR), Success weighted by Path Length (SPL), and Navigation Error (NE, in meters) for navigation quality and efficiency, along with Navigation Step Count (NSC) and Dialog Turn Count (DTC) to capture exploration behavior and dialog efficacy. In addition, we report holistic Localization Error (LE, in meters), which averages localization error over all dialog turns within an episode, providing a measure of how reliably the Guide can infer the Navigator's position throughout the interaction.

## 5.2 RESULTS

**Holistic performance across adopted methods.** Table 1 shows effect of each method we propose. While dataset augmentation alone the effect is marginal (ln1,2), performance improves substantially combined with other methods. With TC, Seen SR jumps from 27.47 to 42.86 (ln2,3), confirming that dialog-aware training better exploits the augmented data. With LC, Unseen SR climbs to 22.82 (ln4), highlighting that stronger localization is particularly beneficial in novel environments. These comparisons indicate that both TC and LC are effective when paired with augmentation. Applying all three components together produces a dramatic boost: Seen SR reaches 58.24 (+29.67 over baseline) and Unseen SR 29.05 (+15.36) (ln5). These results demonstrate that DA, TC, and LC are complementary and, when integrated, deliver strong gains well beyond the baseline.

**Dataset augmentation ablations.** Table 2 analyzes the effect of each stage in our augmentation pipeline. All submodules are trained with the same data variant. Since simply adding fine-grained instruction and trajectory concatenation does not include dialog dataset, only Navigation module is trained with the data variant. Simply adding *fine-grained instructions* provides modest gains (Seen SR +3.3, Unseen SR +2.5)(ln1,2), showing that the richer trajectory–instruction pairs in fine-grained data improve navigation generalization. *Trajectory concatenation*(ln3) extends paths into

Table 3: **Navigation training methods ablation.** + MSB: Model State Building with History. +DET: Decoupled Episodic Training. Augmented dataset from §4.1 and our new localization model (§4.2.2 )are both adopted across all variants. Both training method combined showed the best results on both seen and unseen splits.

| Training Method | Val Seen | | | | | | Val Unseen | | | | | |
|---|---|---|---|---|---|---|---|---|---|---|---|---|
| | SR↑ | OSR↑ | SPL↑ | NE↓ | NSC | DTC | SR↑ | OSR↑ | SPL↑ | NE↓ | NSC | DTC |
| Baseline | 32.97 | 47.25 | 30.76 | 9.26 | 19.85 | 5.76 | 22.82 | 27.80 | 17.00 | 13.18 | 20.89 | 10.57 |
| + MSB | 39.56 | 51.65 | 34.26 | 7.32 | 23.74 | 5.93 | 24.90 | 37.76 | 16.47 | 13.34 | 29.83 | 18.98 |
| + DET | 52.75 | **68.13** | 46.00 | 6.72 | 22.58 | 7.25 | 28.22 | **42.32** | 18.17 | **12.82** | 24.91 | 11.87 |
| **Both (Ours)** | **58.24** | **68.13** | **51.65** | **5.38** | 21.30 | 4.66 | **29.05** | 31.54 | **19.61** | 13.91 | 21.46 | 11.08 |

Table 4: **Localization model ablation.** Segment localization evaluates performance on each question from the RAIN dataset, while holistic localization reports the average accuracy across all dialog turns within an episode. A@0 and A@3 indicate the localization success rate within 0m and 3m of the ground-truth position, respectively.

| | Val Seen | | | | | Val Unseen | | | | |
|---|---|---|---|---|---|---|---|---|---|---|
| | Segment | | | Holistic | | Segment | | | Holistic | |
| | LE↓ | A@0↑ | A@3↑ | SR↑ | LE↓ | LE↓ | A@0↑ | A@3↑ | SR↑ | LE↓ |
| Baseline | 14.14 | 6.63 | 29.20 | 42.86 | 16.51 | 13.30 | 6.61 | 25.33 | 19.92 | 21.45 |
| **Ours** | **8.27** | **23.89** | **46.46** | **58.24** | **5.51** | **10.86** | **12.28** | **34.59** | **29.05** | **13.07** |

multi-turn trajectories, but the improvement is limited, likely due to the constraints of concatenation under-utilizing the available fine-grained instructions. Introducing *dialog turns*(ln4) leads to a substantial jump in performance, with Seen SR reaching 56.04, which is gained from generating question–answer units enabling training for all modules (navigation, QA, and localization). Finally, applying our *reformatting*(ln5) step to convert raw QA pairs into natural multi-turn dialogs yields the best results: Seen SR 58.24 and Unseen SR 29.05. Compared to dialog turns alone, reformatting adds +2.2 SR on Seen and +6.2 SR on Unseen, indicating that natural dialog style not only enhances in-domain efficiency but also strengthens generalization to novel environments. Overall, these results show that while raw fine-grained or concatenated data provide limited benefits, structuring and naturalizing them into dialog form is crucial for unlocking their potential.

**Navigation training ablations.** Table 3 evaluates the effect of our two training strategies: Model State Building (MSB) and Decoupled Episodic Training (DET). Compared to the baseline MSB modestly improves both splits (Seen SR 39.56, Unseen SR 24.90) (ln 1, 2) indicating benefits from conditioning on dialog–navigation history. DE(ln3)T yields a larger jump (Seen SR 52.75, Unseen SR 28.22), confirming that episodic rollouts helps better reflect interactive dynamics. When combined, MSB+DET achieves the best performance (Seen SR 58.24, Unseen SR 29.05), demonstrating their complementary nature: MSB strengthens history modeling while DET stabilizes trajectory learning.

**Localization model ablations.** Table 4 evaluates both *segment-level* localization, which measures accuracy for each individual dialog turn, and *holistic* localization, which aggregates performance across an entire episode. All navigator training schemes (§4.2.1) and dataset augmentations (§4.1) are consistently applied to both setups. Compared to the baseline GCN, our graph-aware transformer shows clear improvements at the segment level across all metrics (A@0, A@3, LE). These improvements carry over to the holistic setting: Success Rate rises both on Seen (42.86 to 58.24) and Unseen (19.92 to 29.05). These results confirm that localization is the primary bottleneck in Dial-Nav and demonstrate that our graph-aware transformer effectively alleviates this limitation, yielding substantial end-to-end improvements.

**Qualitative Analysis** In Figure 4, we provide qualitative comparisons between the baseline Navigator and Guide agent pairs (Han et al., 2025) and ours on the same task instances. While baseline often produces inconsistent responses that mislead localization and result in navigation failure, our model accurately grounds the navigator's position and guides successful navigation.

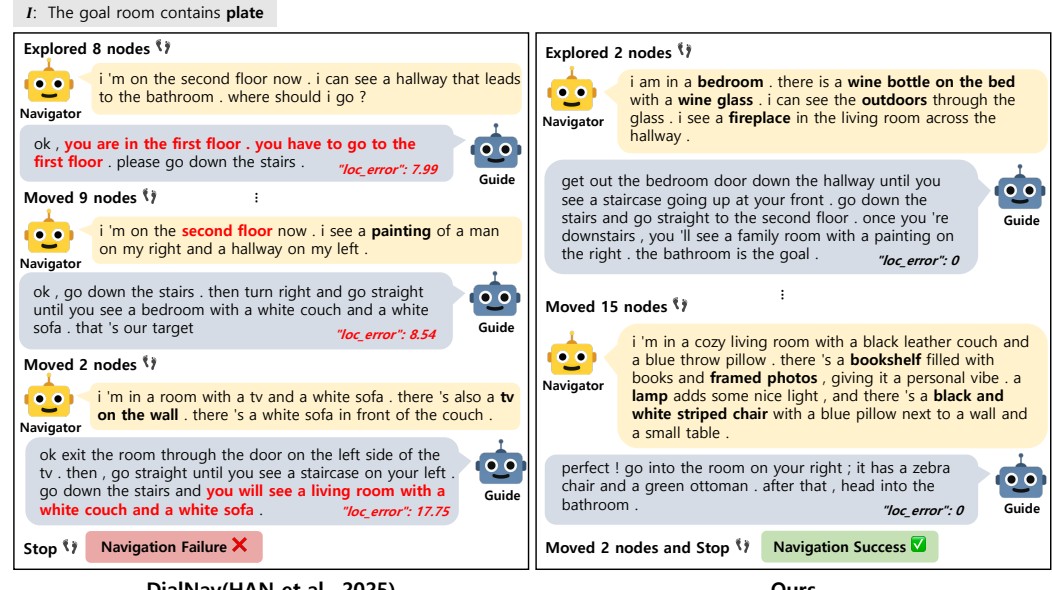

Figure 4: **Qualitative comparisons on the same task instance between the DialNav and Ours.** Compared to DialNav, which produces shorter and less grounded responses leading to failure, our approach generates richer and more precise language that remains consistent with the environment and guides the Navigator to success. Full examples with viewpoint images are provided in Supp. Mat. G.

## 6 CONCLUSION

In this work, we addressed the limitations of DialNav by advancing data, training, and modeling toward a holistic solution for dialog-based navigation. First, we introduced a large-scale automatic dialog generation pipeline that expands the limited RAIN dataset by transforming static VLN trajectories into coherent multi-turn dialogs, significantly increasing training coverage and diversity. Second, we proposed dialog-aware navigator training, combining history rollout and episodic supervision to better align learning with the dynamic dialog–navigation loop. Finally, we highlighted localization as the critical bottleneck and adapted DUET into a graph-aware localizer, achieving substantial improvements in grounding Navigator questions to environment nodes. Extensive experiments demonstrated that these advances markedly improve both localization accuracy and navigation success, substantially narrowing the gap to oracle performance. Beyond providing stronger baselines, our pipeline and training strategies establish a scalable foundation for future research in dialog-driven navigation, where agents must operate robustly in realistic, interactive embodied settings.

## ETHICS STATEMENT

We acknowledge that LLMs were used as writing assistants to improve grammar, clarity, and readability of the manuscript.

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

## A   DATASET STATISTICS

We compare the original **RAIN** dataset (2,233 episodes) with our automatically generated **Rainbow** dataset (238,028 episodes). Table 5 reports mean, median, maximum, and minimum values across episodes for key statistics.

Table 5: **Comparison of dataset statistics between RAIN and Rainbow.** Rainbow is more than two orders of magnitude larger, with richer dialogs and more diverse language.

| Statistic | RAIN | | | | Rainbow (Ours) | | | |
|---|---|---|---|---|---|---|---|---|
| | Mean | Median | Max | Min | Mean | Median | Max | Min |
| # Episodes | | 2,231 | | | | 238,028 | | |
| Dialog turns / ep. | 1.88 | 2 | 8 | 0 | 2.71 | 3 | 4 | 1 |
| Question length (words) | 27.48 | 25 | 119 | 1 | 30.43 | 29 | 125 | 4 |
| Answer length (words) | 42.53 | 38 | 184 | 1 | 43.17 | 36 | 300 | 0 |
| Trajectory length (nodes) | 26.03 | 23 | 116 | 4 | 19.54 | 19 | 67 | 2 |

## B   QUESTION AND ANSWER GENERATION MODEL

For both question and answer generation, we adopt the **LANA** model (Wang et al., 2023a) as our baseline module. LANA is a vision-and-language navigation (VLN) model originally designed to enhance navigation performance by jointly learning to describe not only the upcoming path but also the trajectory history that has already been traversed. This ability to ground language in both past This makes LANA naturally aligned with the dialog setting in DialNav, where generated utterances must be grounded in the environment.

In our setup, LANA is initialized with pretrained weights and fine-tuned on the RAIN dataset to adapt navigation-style instructions into dialog-style questions and answers. For *question generation*, the model takes as input the panoramic observation at the Navigator's current viewpoint and outputs a natural language question describing salient local features, enabling the Guide to localize the Navigator. For *answer generation*, the model receives as input a sequence of panoramic images corresponding to the next subpath toward the goal region, and outputs a natural language description of this subpath as a guiding response. By grounding both questions and answers in visual observations and trajectory context, LANA provides a consistent and suitable baseline for DialNav dialog modeling.

## C   LOCALIZATION MODEL

For localization, we benchmark a lightweight cross-modal architecture, the **Graph Convolutional Network (GCN) Localization** Hahn & Rehg (2022), which operates directly on the Matterport3D navigation graph and formulates localization as a *node-selection* problem. Given the Guide's last question $q$, panoramic visual descriptors $\{x_v\}_{v \in V}$ at all nodes, and the graph topology $G = (V, E)$, the model first fuses language and vision features at each node. These representations are then propagated across the graph via message passing, enabling each node embedding to incorporate contextual information from its neighbors. A per-node classifier outputs a distribution $p(v \mid q, G)$ over candidate nodes, and the ground-truth node is supervised with cross-entropy loss.

After *Where Are You? (WAY)* (Hahn et al., 2020) introduced embodied localization as an independent research problem, subsequent studies have been proposed. However these works (Wang et al., 2025; Zhang et al., 2024; Pate & Wong, 2023; Li et al., 2024) largely focused on predicting the agent's pose in a *top-down* view, often regressing continuous coordinates or grid cells with access to bird's-eye maps. While conceptually related, such approaches differ from our setting, which formulates localization as discrete node selection on the navigation graph. Following prior work on DialNav, we therefore constrain our baselines to graph-based localization models. However, several state-of-the-art graph-based systems (Hahn & Rehg, 2022) do not release code or pretrained weights, preventing controlled comparisons in our setup.

Consequently, we adopt GCN as a reproducible baseline for node-level localization, and beyond this, introduce our DUET-based, graph-aware localizer tailored to the holistic DialNav setup.

## D    TRAJECTORY CONCATENATION DETAILS

For trajectory concatenation, we applied the following constraints to ensure natural continuity:

1. We concatenated 2–4 trajectories from R2R Anderson et al. (2018), RxR Ku et al. (2020), CVDN Thomason et al. (2020) answer trajectories into a single episode.

2. The endpoint of one trajectory and the start of the next must be within 1 meter in the navigation graph.

3. To prevent overly circuitous paths, the detour ratio—the concatenated path length divided by the shortest path length between the start and end nodes—was constrained to be less than 1.3.

4. Episodes in which the goal region contained no selectable object were discarded.

5. The ambiguous instruction $I$ was derived from the Matterport3D Chang et al. (2017) metadata by randomly selecting one visible object in the goal region. To avoid trivial or overly generic references, we excluded a predefined set of categories (e.g., `wall`, `floor`, `ceiling`, etc.).

6. Since goal regions in DialNav correspond to rooms rather than single nodes, we excluded cases where the agent had already reached the goal room before subsequent dialog turns were appended, avoiding unnatural "post-goal" interactions.

7. To further increase diversity, we additionally introduced variations in 10% of the constructed episodes, simulating potential deviations in real dialog navigation. We consider three types of variations: *mislocalization*, *misnavigation*, and *exploration*. In the case of mislocalization, the Guide intentionally provides an incorrect path description that does not match the Navigator's true position; the Navigator then proceeds by moving 1–2 nodes randomly from the original location rather than following the erroneous instruction. For misnavigation, the Navigator deviates from the instructed path and follows a randomly chosen alternative route. In exploration, the Navigator continues beyond the instructed trajectory by taking an additional 2–5 random steps after completing the suggested path. By incorporating these cases, we were able to include $N$ additional instructions that were previously filtered out by the strict trajectory-connection criteria, thereby maximizing the utilization of existing VLN data for dialog augmentation.

## E    CAPTION GENERATION DETAILS

To produce natural and visually grounded questions $q_i$ at each dialog node $v_i$, we prompted LLaVA-1.5-7B and LLaMA-3.1-8B with different visual–textual contexts. Our aim was to ensure that the resulting utterances contained sufficient local detail for Guide-side localization. We experimented with three variants:

**(A) Simple panoramic caption.**    The panoramic observation at $v_i$ is directly given to LLaVA with a short instruction to generate a navigation-oriented question. This produces concise, layout-focused questions. The corresponding prompt is shown in Fig. 5 (top).

**(B) Detailed panoramic caption.**    Here, we instructed LLaVA to provide a more elaborate description of the panoramic scene, explicitly mentioning the room type as well as distinctive objects or layout features. This yields richer, more descriptive questions. The prompt is provided in Fig. 5 (middle).

**(C) Object-grounded panoramic caption.**    Using Matterport3D metadata, we extract the set of visible objects at $v_i$. For each object, its image patch is cropped from the panorama, and a caption is generated with LLaVA. We then jointly supply (i) the panoramic caption, (ii) a randomly sampled subset (50% or 70%) of object-level captions, and (iii) the room type, prompting LLaMA to generate a single detailed caption that integrates both global layout and salient object cues. To

increase diversity, we vary the sampling ratio and the number of in-context examples (1/3/5-shot). The prompt is provided in Fig. 5 (bottom).

Samples of generated captions from these variants are shown in Fig. 6.

## F    DIALOG REFORMAT DETAILS

The raw dialog data collected from automatic pipelines often contains instruction-like or overly verbose utterances, which are less natural than human conversations. To address this, we reformatted all dialogs into a more conversational style while preserving their semantic intent. The process consisted of three steps:

1. **Goal-conditioned branching.** For each dialog episode, we generated two versions of every utterance: one that explicitly mentions the goal (e.g., "You have reached your destination"), and one that does not. During trajectory concatenation, we used the goal-oriented variant only for the final turn, while the intermediate turns relied on the neutral version. This explicit branching strategy prevents ambiguous or inconsistent goal mentions. The exact instructions for this step are given in the goal-conditioned reformat prompt (Figure 7).

2. **Dialog smoothing.** The full dialog was then paraphrased by GPT-4o-mini into natural, conversational language while retaining all navigation-critical details. Overly formal or cumbersome phrases were simplified, redundant expressions were removed, and subtle acknowledgments were inserted where appropriate to enhance conversational flow. We used the natural dialog reformat prompt (Figure 8) to guide this step.

Figure 9 presents qualitative comparisons of original (instruction-style) versus reformatted dialogs across multiple samples. The reformatting improves dialog naturalness, reduces redundancy, and better reflects realistic Navigator–Guide interactions.

## G    QUALITATIVE EXAMPLES

Figure 11 presents qualitative comparisons on the same task instance between the DialNav Han et al. (2025) and our model. The DialNav Guide fails in localization of the Navigator and ultimately describes an incorrect route, which results in navigation failure. In contrast, our model enables the Navigator to accurately describe its position, allowing the Guide to successfully localize and provide appropriate guidance, leading to successful navigation.

**Prompt for Simple Panoramic Caption**

```
Generate a sentence that describes the given panoramic image.  Start your question with
"I'm in ..." or "I can see ...".
```

**Prompt for Detailed Panoramic Caption**

```
You will be given a panoramic image of an indoor scene.  Describe the scene in detail:
state the type of room (e.g., bedroom, bathroom, hallway, stair, empty room) and
mention distinctive objects, furniture, or layout features that uniquely identify
this room compared to other rooms in a building.  The description should highlight both
the function of the room and its unique elements.  Start your description with "The
[room type] in the image is ...".
```

**Prompt for Object-grounded Panoramic Caption**

```
You are provided with (1) a panoramic caption that describes the overall layout and
appearance, (2) a list of visible objects denoted as <OBJECT_LIST>, and (3) the room
type indicating the semantic category of the space.  Based on this information,
generate a single natural sentence that clearly and concisely describes the scene
in a way that would help another person understand your exact surroundings.  Begin your
response with "I'm in ..." or "I can see ...".
```

Figure 5: Prompt templates used for generating navigation questions from panoramic observations. We employ three variants: simple panoramic caption (top), detailed panoramic caption (middle), and object-grounded panoramic caption (bottom).

| Simple caption | *"I'm in a living room with a wooden floor."* |
|---|---|
| **Detailed caption** | *"The kitchen in this image is a small, cozy space with a dining table and chairs. The table is surrounded by chairs, and there is a potted plant nearby. The kitchen is part of a log cabin, which adds to the rustic and warm atmosphere of the space."* |
| **Object-grounded caption** | |
| 1-shot, 50% objects | *"I'm in a cozy log cabin kitchen. The room has a blue wooden ceiling and walls, giving it a rustic and warm atmosphere. There's a dining table with chairs surrounding it, and a potted plant nearby. The walls are decorated with a clock, a vase, and a bowl, adding to the inviting ambiance."* |
| 1-shot, 70% objects | *"I'm in a cozy log cabin kitchen. The room is small and warm, with a rustic atmosphere. There's a dining table and chairs in the center of the room, surrounded by wooden walls and a blue wooden ceiling. A potted plant is nearby, adding a touch of greenery to the space."* |
| 3-shot, 50% objects | *"I'm in a small, cozy kitchen inside a log cabin. The room has a rustic and warm atmosphere, with a blue wooden ceiling and walls. There's a dining table with chairs surrounding it, and a potted plant nearby."* |
| 3-shot, 70% objects | *"I'm in a cozy log cabin kitchen. The room has a rustic feel with a wooden ceiling and walls. There's a small dining table with chairs surrounding it, and a potted plant nearby. The atmosphere is warm and inviting."* |
| 5-shot, 50% objects | *"I'm in a cozy log cabin kitchen. The room has a blue wooden ceiling and walls, giving it a rustic and warm atmosphere. There's a dining table with chairs surrounding it, and a potted plant nearby. The walls are decorated with various items, including a clock, a vase, and a bowl."* |
| 5-shot, 70% objects | *"I'm in a small, cozy kitchen that's part of a log cabin. The room has a rustic and warm atmosphere, with a blue wooden ceiling and walls. There's a dining table with chairs surrounding it, and a potted plant nearby."* |

Figure 6: Diversity of generated captions.

---

**Prompt for Goal-conditioned Reformat**

```
You will be provided with a sequence of sentences describing a path.
Your task is to rewrite the path description in two versions:
1. with_goal: A version that includes reaching the final destination
2. without_goal: A version that excludes any mention of reaching the destination
Both versions should:
- Be concise while preserving key spatial references
- Focus on objects, rooms, and layout
- Use directions only when necessary
- Remove terms implying current situation (Start facing , You're in a , You're
facing)
For with_goal version, Always mention that you've arrived at the destination at the end
of the description with expressions like: That's the goal, You have arrived, That's
your destination, The goal, You've reached your destination, This is the final spot,
You're at the destination, Mission accomplished, End point reached, You've made it.
For without_goal version, Remove any mentions of arriving at or reaching the final
destination.
```

Figure 7: Prompt template for goal-conditioned path reformatting.

864
865
866
867
868
869
870
871
872
873
874
875
876
877
878
879
880
881
882
883
884
885
886
887
888
889
890
891
892
893
894
895
896
897
898
899
900
901
902
903
904
905
906
907
908
909
910
911
912
913
914
915
916
917

## Prompt for Natural Dialog Reformat

```
You will rewrite a dialog between a Navigator and a Guide in an indoor navigation task.
TASK CONTEXT:
- The Navigator and Guide work together to reach a goal room.
- Both Navigator and Guide know about a shared object in the goal room.
- The Guide knows the goal room but the Navigator does not.
- The object may appear in other rooms, so the Navigator must ask clarifying questions.
- The Guide cannot see the Navigator's position but knows the environment layout.
- Dialog alternates turns:  Navigator → Guide → Navigator ...  (no limit on turns).
- Dialog always starts with the Navigator.
OBJECTIVE:
- Rephrase the dialog to sound like natural conversation.
- Style:
* Use casual, friendly language ("Ok", "Alright", "Got it").
* Add acknowledgments ("I see", "Understood").
* Smooth transitions between turns.
* Vary vocabulary, keep meaning the same.
- Actively incorporate mentions of the shared object in the conversation.
- The Guide's final response should mention reaching the goal room.
RULES:
1.  Keep the same number of turns and same speakers (Navigator/Guide).
2.  Do not add or remove turns.
3.  Do not add new objects, rooms, or details not in input.
4.  Never use phrases like "You're looking at..." or "You are facing...".  - Instead:
ask questions ("Do you see...?") or omit.
5.  Do not drop or shorten navigation instructions.
6.  Preserve all key details:
- Objects and attributes (color, shape, material).
- Room names and types.
- Spatial relations and directions.
7.  Do not use em dashes (|).  Use commas, periods, or "and".
OUTPUT FORMAT:
Return ONLY valid JSON:
"reformatted":  [
"Navigator":  "...", "Guide":  "..." ,
"Navigator":  "...", "Guide":  "..."
]
No extra text, no markdown, no explanations.
```

Figure 8: Prompt template for natural dialog reformatting.

---

**Example**

---

**Original Instruction**

*"We start off looking at the corner of a wall. As you turn around you see a large open are with doorways to many rooms around you. On one of the walls you see a white rectangular box. Take a step towards that white rectangular shape and turn to look down the hall and you will see two chairs directly in front of you down the hall. And before those two chairs on the right there is an open walkway. Take a step towards that open walkway on your right. You will be looking into a room that has brown cabinets and a shelf with a white pillow and radio looking thing on it **and you are done.**"*

**Reformatted (without_goal)**

*"From the corner of a wall, turn to see a large open area with doorways to various rooms. Approach the white rectangular box on the wall, then look down the hall to see two chairs ahead. To the right of the chairs, there's an open walkway. Step into the walkway to view a room with brown cabinets and a shelf containing a white pillow and a radio."*

**Reformatted (with_goal)**

*"Turn away from the wall to find a large open area with doorways to various rooms. Move toward the white rectangular box on the wall, then look down the hall to see two chairs ahead. To the right of those chairs is an open walkway; step into it. You'll see a room with brown cabinets and a shelf containing a white pillow and a radio. **You've reached your destination.**"*

---

Figure 9: **Examples of goal-conditioned branching.** Each example shows the original instruction followed by two reformatted variants: *without_goal* and *with_goal*.

| | Before Reformat | After Reformat |
|---|---|---|
| **N1** | I'm in a long hallway filled with books from floor to ceiling. The bookshelf stretches the entire length of the hallway, packed with books of all sizes and types, creating an impressive display of knowledge and resources. | I'm in this long hallway packed with books all the way up to the ceiling. The bookshelf runs the entire length of the hallway, filled with all sorts of books, creating an amazing sight. |
| **G1** | You are standing in a library, to the right you have a book shelf, turn back from the place where you are standing, walk straight, till the end, turn right, walk straight, until you reach the book shelf in-front of you, turn right, walk straight, walk straight until you reach the book shelf in-front of you and a staircase on your right, get down the stairs completely. Slightly turn right, in-front of you there is a large table with chairs around it, to the right you have a book shelf, stand between the table and book shelf, turn back. Sorry, turn right, in-front of you there is an open door walk to the open door, in-front of you there is a staircase, stand on the staircase, that's your end point. | Awesome! In the library, head straight to the end and take a right. Keep going until you reach another bookshelf, then make another right and head toward the bookshelf with a staircase on its right. Go down the staircase completely, and then turn a bit right to find a big table surrounded by chairs and another bookshelf on the right—stand right between those two. Next, turn right toward the open door and walk through it to reach the staircase. |
| **N2** | The library in this image is a room with a stone archway and a white wall. The room appears to be a part of a university, as it is described as a "college library." The stone archway adds a unique architectural element to the space, making it visually appealing. The white wall complements the stone archway, creating a clean and elegant atmosphere. The room is likely used for studying, research, and quiet contemplation, as it is a library setting. | I see the library has a stone archway and a white wall. It feels like a university space, really charming. |
| **G2** | You are now facing the white color door, from there take a left turn. Start going down the stairs and stand on the tenth step, you will find your destination point. | Perfect! Now, turn left from the white door and go down the stairs until you hit the tenth step. That's your goal. |

| | Before Reformat | After Reformat |
|---|---|---|
| **N1** | I am in a large, wooden wardrobe with a mirrored door. The closet is filled with various clothing items, including shirts and suits, and is well-organized, making it easy to find and access the clothes stored inside. | I'm inside this large wooden wardrobe with a mirrored door. It's packed with all kinds of clothes like shirts and suits, and everything is organized really well, so it's easy to find what I need. |
| **G1** | Walk down the hallway to your right and move past the bed. Walk through the door on the right of the painting of thr girl and wait there. | Alright! Start by heading down the hallway to your right and walk past the bed. Then, go through the door on the right side of the girl's painting and just wait there. |
| **N2** | I'm in a curved hallway with a staircase and glass railing, surrounded by various artworks on the walls, including paintings and photographs. | I've made it to a curved hallway that has a staircase and a glass railing. The walls are decorated with various artworks, including paintings and photos. |
| **G2** | down more stairs, and into the little room left of the silver mirrored planter | Nice! Now, head down those stairs and enter the small room to the left of the silver mirrored planter. You've finally arrived at your destination. |

| | Before Reformat | After Reformat |
|---|---|---|
| **N1** | The bathroom in the image features a toilet and a sink. The toilet is positioned on the right side of the bathroom, while the sink is located on the left side. The bathroom also has a window, which allows natural light to enter the space. The overall design of the bathroom is simple and functional, with a focus on the essential fixtures. | I've got a bathroom here, it's got a toilet on the right and a sink on the left. There's a window letting in some natural light too. The design is pretty simple but works well. |
| **G1** | You're in a bathroom, turn to the right and walk through the doorway. Walk over the carpet and to the colorful stairs. Walk up a step and then wait. | I see! Now, turn to your right and head through the doorway. Walk over the carpet until you reach the colorful stairs. Once you're up a step, just hang tight for a moment. |
| **N2** | I'm in a hallway with a staircase and a doorway. | I'm now in a hallway with a staircase and a doorway. |
| **G2** | Go to the top of the stairs until you see two chairs and one table. Following that, make a right towards the bedroom and wait there. | Perfect! Continue to the top of the stairs where you'll spot two chairs and a table. After that, make a right into the bedroom and just wait there. You've made it to your goal! |

Figure 10: Side-by-side comparison of original vs. reformatted dialogs for three separate samples (each table). (**N** = Navigator, **G** = Guide). Red text in the left columns highlights less conversational or cumbersome phrases; blue text in the right columns highlights improved conversational tone.

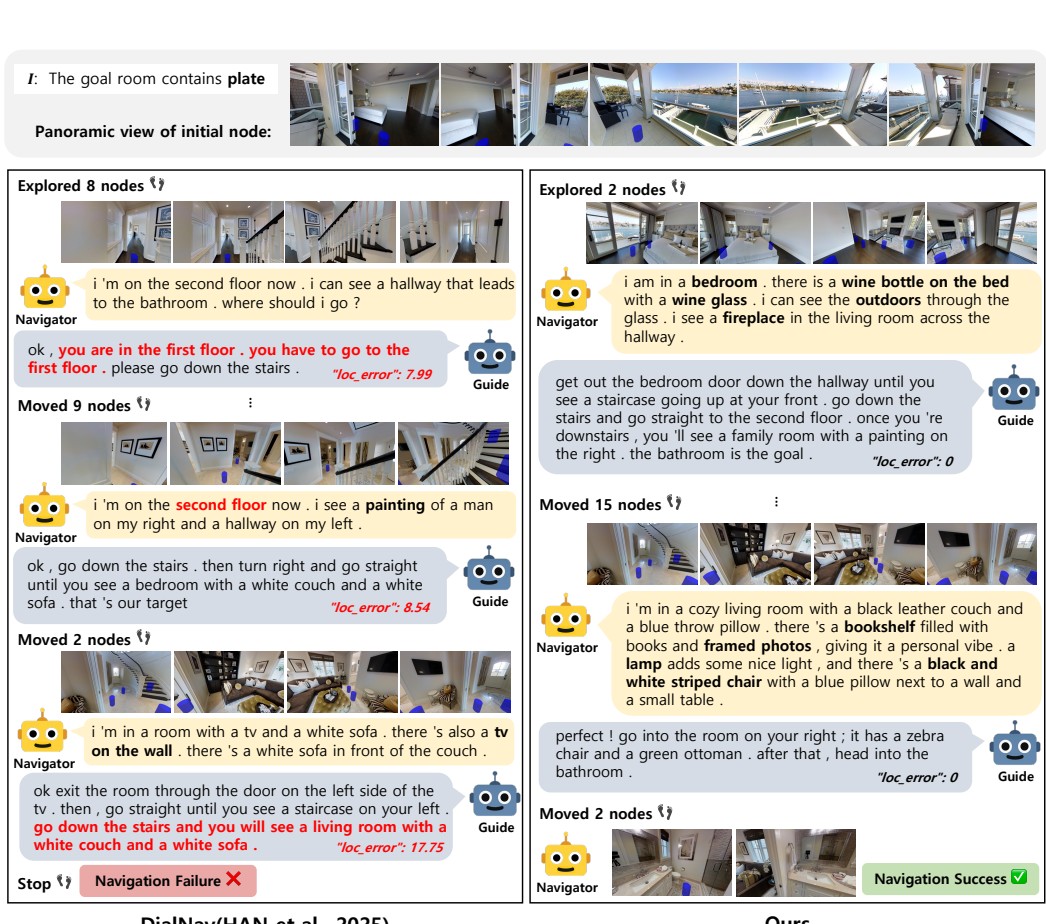

Figure 11: **Qualitative comparisons on the same task instance between the DialNav and Ours.**

