# OpenReview forum: "Holistic Advances through Large-Scale Embodied Dialog Augmentation for Navigation"
_ICLR.cc/2026/Conference — ICLR 2026 Conference Withdrawn Submission_

### Official Review · Reviewer_6X4A · 2025-10-16

**Soundness:** 3
**Presentation:** 3
**Contribution:** 2
**Rating:** 4
**Confidence:** 4

**Summary:**

This paper tackles three key bottlenecks in dialogue-based vision-and-language navigation (DialNav): limited training data, static training paradigms, and inaccurate localization. The authors propose a holistic solution that (1) automatically generates large-scale multi-turn navigation dialogues by reformulating existing VLN trajectories using LLMs, (2) introduces interaction-aware navigator training with history-conditioned rollouts and episodic supervision, and (3) replaces the baseline localizer with a graph-aware transformer adapted from DUET. Their full system achieves substantial improvements and significantly reduces localization error.

**Strengths:**

- **Comprehensive ablation study**: The paper systematically evaluates the impact of data augmentation, trajectory concatenation, dialog-aware training, and localization modeling, clearly isolating the contribution of each component.
- **Effective large-scale dialog synthesis**: The LLM-based pipeline produces fluent, visually grounded dialogues that significantly expand training coverage beyond the original RAIN dataset, enabling better generalization.
- **Clear presentation**: The paper is well-written, with a logical flow and precise technical descriptions. Key concepts, such as dialog reformulation and interaction-aware rollouts, are effectively illustrated through intuitive figures, enhancing readability.

**Weaknesses:**

- **Limited algorithmic novelty**: While the integration is effective, each technical component—LLM-based data generation, history-aware training, graph-based localization—is an adaptation of existing ideas from VLN or dialogue systems. The work advances engineering practice but lacks a new conceptual or architectural insight.
- **Unclear distinction from recent data augmentation approaches**: The paper does not sufficiently differentiate its dialog reformulation strategy from prior VLN data augmentation methods (e.g., NavRAG, VLN-SRDF). What is the essential difference of them? Without this analysis, it’s hard to assess whether gains stem from the pipeline design or simply scale.
- **Narrow evaluation scope**: Experiments are confined to the RAIN benchmark, which uses room-level goals and synthetic dialogues. The approach is not evaluated on other ealistic embodied dialog navigation benchmarks such as CVDN, limiting the claimed broader impact.
- **Lack of computational overhead analysis**: While the overall framework is compelling, the paper lacks a discussion of the computation overhead. Given this work utilizes , GPT-4o-mini for data process, it would be valuable to understand the associated computational cost.

**Questions:**

See Weaknesses

---

### Official Review · Reviewer_jJ6z · 2025-10-31

**Soundness:** 3
**Presentation:** 3
**Contribution:** 3
**Rating:** 6
**Confidence:** 5

**Summary:**

This paper presents incremental results on the embodied dialogue navigation task.

**Strengths:**

The paper compiles a set of novel methods to improve the task. This includes dataset augmentation, new training methodology and localization techniques. They have achieved significant improvements on the task, and performed reasonable ablations.

**Weaknesses:**

The biggest concern is that the paper still reads like a pre-LLM era paper.
While DUET is transformer based, it is not a SOTA generative VLM such as GPT-5 or Qwen-VL.
I wonder how they would have performed in your task, or how you can apply your new methods using these VLMs.
Couple more papers to cite:
1. Just ask: An interactive learning framework for vision and language navigation
2. Do As I Can, Not As I Say: Grounding Language in Robotic Affordances

**Questions:**

No questions

---

### Official Review · Reviewer_xNnV · 2025-11-03

**Soundness:** 2
**Presentation:** 3
**Contribution:** 2
**Rating:** 4
**Confidence:** 4

**Summary:**

The paper builds a large, synthetic dialogue dataset for VLN by converting existing VLN paths into multi-turn, grounded conversations. It then trains agents with two procedures: Model State Building (to encode dialogue history) and Decoupled Episodic Training (to align rollouts with dialogue), and swaps a GCN localizer for a transformer to reduce cascading localization errors. On the RAIN benchmark, the model obtains gains over prior baselines on both seen and unseen splits.

**Strengths:**

1. The paper provides a scalable data generation pipeline to synthesize a dialogue dataset for VLN.
2. The paper provides a comprehensive ablation of the effectiveness of the proposed methods.
3. The paper is well-written and easy to follow.

**Weaknesses:**

1. Results are largely in-distribution (RAIN). Please add cross-dataset transfer (e.g., to other dialog-navigation benchmarks) and scaling studies showing whether transfer improves as the size of the synthetic corpus grows.
2. Gains concentrate on Val-Seen, while Val-Unseen remains low (SR=29.5%). This suggests the synthesis primarily boosts in-distribution fit. Providing analyses disentangling overfitting vs. robust gains will strengthen the contribution.
3. Lack of error analysis. Include qualitative case studies and a clear failure taxonomy for Unseen. What capabilities improve with more data, and what are the typical failure modes?
4. The system does not leverage the latest LLM/VLMs. Can this dialog–navigation synthetic dataset be used to further train or instruction-tune general VLMs to systematically improve VLN understanding and planning? Could it serve as an intermediate task (e.g., visual instruction-following or multi-turn alignment) that yields measurable transfer gains?
5. The intro motivates dialog for safety, but the experiments don’t measure safety-relevant outcomes

**Questions:**

1. How do you guard against hallucinated or spatially inconsistent LLM rewrites? Any automatic or human validation of dialog consistency with panoramas and paths?
2. What happens if RAIN:Aug is 1:1 or 1:3? Does Unseen SR peak near 1:9?

---

### Official Review · Reviewer_FDEA · 2025-11-04

**Soundness:** 3
**Presentation:** 3
**Contribution:** 2
**Rating:** 4
**Confidence:** 3

**Summary:**

The paper tackles the limitations of current dialog-based embodied navigation (DialNav) methods, which struggle with small datasets, weak dialog–navigation coupling, and poor localization. To address this, they propose a large-scale data generation pipeline that uses LLMs and VLM models to synthesize realistic multi-turn navigation dialogs from existing VLN datasets. They further propose two training strategies: Model State Building (MSB), which enables the agent to accumulate dialog and navigation context before acting, and Decoupled Episodic Training (DET), which separates rollout and supervision to improve learning stability. Finally, they incorporate a graph-aware transformer localizer to enhance spatial grounding. The combination of these components yields strong quantitative improvements on the DialNav benchmark.

**Strengths:**

### 1. Large-scale dataset construction:
The pipeline that leverages LLMs to generate dialog data from existing VLN datasets is practical and valuable. The resulting dataset expands the original RAIN benchmark (~2K episodes) to over 230K dialog–navigation trajectories, providing a substantial scale-up. This extra data would help to scale VLN methods to more diverse and realistic dialog scenarios.

### 2. Strong empirical performance:
The model substantially outperforms the baseline on the DialNav benchmark, achieving nearly double the success rate on both seen and unseen splits.
### 3. Ablations:
Through ablations, they show that data augmentation, using dialog history as context, and incorporating the localization module each contribute meaningfully to the overall performance.

**Weaknesses:**

### Limited conceptual novelty:
The proposed methods (MSB, DET, and the graph-aware localizer) largely repackage established ideas from prior work.
- MSB closely resembles memory- or history-aware models such as HAMT [1] and MTVM [2], which already model long-horizon visual and linguistic context for navigation.
- DET resembles standard episodic training and rollout-decoupling techniques used in imitation and reinforcement learning, as seen in EnvDrop [3] and Recurrent VLN-BERT [4], without explanation of how the proposed method is different and without clear theoretical or algorithmic distinction.
- The graph-aware transformer localizer directly uses DUET [5] which is a graph-transformer model for navigation grounding. Moreover, localization from dialog has already been explored by LED [6], which uses transformer-based graph localization from dialog input. The paper does not adequately differentiate these from its own modules nor provide substantial new insights.

### Dataset scaling is incremental, not a core research contribution:
The “Rainbow” dataset is generated automatically using existing data and off-the-shelf LLMs (LLaVA, GPT-4o-mini). While useful for the community, this is primarily an engineering contribution rather than a conceptual one. The authors’ pipeline, while effective, does not demonstrate fundamental innovation in data generation methodology or modeling.


### References

[1] History Aware Multimodal Transformer for Vision-and-Language Navigation (HAMT), NeurIPS 2021.

[2] Multimodal Transformer with Variable-Length Memory for Vision-and-Language Navigation (MTVM), ECCV 2022.

[3] Learning to Navigate Unseen Environments: Back Translation with Environmental Dropout, NAACL 2019.

[4] Recurrent Vision-and-Language BERT for Navigation (Recurrent VLN-BERT), CVPR 2021.

[5] Chen, S., Guhur, P.-L., Tapaswi, M., Schmid, C., Laptev, I. Think Global, Act Local: Dual-Scale Graph Transformer for Vision-and-Language Navigation (DUET), CVPR 2022.

[6] Transformer-based Localization from Embodied Dialog (LED), AACL 2022.

**Questions:**

-	How is MSB functionally distinct from existing memory-based or history-aware transformers such as HAMT, MVTM?
-	What is fundamentally new about DET beyond standard decoupled rollout or episodic training used in reinforcement learning beyond proposed in EnvDrop, Recurrent-VLNBERT?
-	How does your graph-aware localizer differ from DUET or LED?

Basically, a careful comparison with past VLN methods is required to show the merits of the overall approach, in addition to the increased data scale.

---

### Note · Authors · 2025-11-14

I have read and agree with the venue's withdrawal policy on behalf of myself and my co-authors.